# Recent Advances on Biomarkers of Early and Late Kidney Graft Dysfunction

**DOI:** 10.3390/ijms21155404

**Published:** 2020-07-29

**Authors:** Marco Quaglia, Guido Merlotti, Gabriele Guglielmetti, Giuseppe Castellano, Vincenzo Cantaluppi

**Affiliations:** 1Nephrology and Kidney Transplantation Unit, Center for Translational Research on Autoimmune and Allergic Disease (CAAD), Department of Translational Medicine, University of Piemonte Orientale (UPO), AOU Maggiore della Carità, via Gen. P. Solaroli, 17-28100 Novara, Italy; marco.quaglia@med.uniupo.it (M.Q.); guido.merlotti@maggioreosp.novara.it (G.M.); g.guglielmetti@maggioreosp.novara.it (G.G.); 2Nephrology, Dialysis and Transplant Unit, Department of Medical and Surgical Sciences, University of Foggia, 71121 Foggia, Italy; giuseppe.castellano@unifg.it

**Keywords:** renal transplant, biomarkers, extracellular vesicles, acute rejection, chronic rejection, chronic allograft dysfunction, calcineurin-inhibitor nephrotoxicity, Polyomavirus associated nephropathy, immunosuppression

## Abstract

New biomarkers of early and late graft dysfunction are needed in renal transplant to improve management of complications and prolong graft survival. A wide range of potential diagnostic and prognostic biomarkers, measured in different biological fluids (serum, plasma, urine) and in renal tissues, have been proposed for post-transplant delayed graft function (DGF), acute rejection (AR), and chronic allograft dysfunction (CAD). This review investigates old and new potential biomarkers for each of these clinical domains, seeking to underline their limits and strengths. OMICs technology has allowed identifying many candidate biomarkers, providing diagnostic and prognostic information at very early stages of pathological processes, such as AR. Donor-derived cell-free DNA (ddcfDNA) and extracellular vesicles (EVs) are further promising tools. Although most of these biomarkers still need to be validated in multiple independent cohorts and standardized, they are paving the way for substantial advances, such as the possibility of accurately predicting risk of DGF before graft is implanted, of making a “molecular” diagnosis of subclinical rejection even before histological lesions develop, or of dissecting etiology of CAD. Identification of “immunoquiescent” or even tolerant patients to guide minimization of immunosuppressive therapy is another area of active research. The parallel progress in imaging techniques, bioinformatics, and artificial intelligence (AI) is helping to fully exploit the wealth of information provided by biomarkers, leading to improved disease nosology of old entities such as transplant glomerulopathy. Prospective studies are needed to assess whether introduction of these new sets of biomarkers into clinical practice could actually reduce the need for renal biopsy, integrate traditional tools, and ultimately improve graft survival compared to current management.

## 1. Introduction

### General Features and Meaning of a Biomarker

A biomarker has been defined as “a characteristic that is objectively measured and evaluated as an indicator of a normal biological process, pathogenic process or pharmacological response to a therapeutic intervention” [1,2].

Transplanted kidney is currently monitored through a complex of clinical (e.g., GFR, proteinuria), immunological (e.g., DSA), instrumental (e.g., resistive index at Doppler ultrasound), and histological parameters. Overall these “traditional biomarkers” have many limits related not only to disease, but also to both nephrologists’ and pathologists’ skills. Even histological examination through renal biopsy, which remains the diagnostic golden standard criterion despite its invasiveness, is hampered by many drawbacks: low sensitivity (e.g., failure to detect subclinical acute rejection), low specificity due to heterogeneity of processes underlying the same lesion (e.g., uncertain interpretation of interstitial fibrosis-tubular atrophy, IFTA), lack of standardization (poor reproducibility, elevated inter-observer variability due to expertise-dependence) and of quantitative thresholds, sampling errors (e.g., failure to detect focal disorders such as Polyomavirus associated nephropathy, PVAN) [3].

New biomarkers have been the focus of intense research over the last decade to overcome these limits and improve allograft monitoring. Most of them are derived from “OMICs” revolution [4] and can be considered the cornerstone of precision medicine, which is based on a proactive approach and aims at predicting and preventing pathological processes by providing earlier and more extensive information than traditional ones [5].

In general, biomarkers can be classified into seven categories with different meaning and aims [5], outlined in Table 1.

A plethora of new, non-invasive biomarkers measured in either urine or peripheral blood have been studied over the last years, mainly with a diagnostic and prognostic meaning, with different degrees of preclinical and clinical success. Some of them have been validated in independent cohorts and may be already employed in clinical decision-making when kidney biopsy is contraindicated or inconclusive. Other biomarkers, mainly represented by gene expression signatures, have been assessed in kidney tissue and appear to significantly expand information provided by traditional histology [6].

Different pathological processes can cause early and late KTx dysfunction and are outlined in Figure 1.

We herein reviewed the current literature on potential biomarkers in three main settings of KTx: ischemia reperfusion injury (IRI) and DGF, AR, and CAD. The latter includes biomarkers for chronic rejection, chronic Calcineurin-Inhibitor (CNI) nephrotoxicity, and PVAN.

## 2. IRI and DGF

DGF is a common complication of KTx, which affects short and long-term outcomes, including risk of acute rejection and graft survival. It is often caused by IRI due to long cold ischemia time, especially in kidney from “extended-criteria” donors (ECD) and donation after cardiac death (DCD). The most commonly employed definition for DGF relies on the need for dialysis in the first week after KTx.

Biomarkers measured in the immediate post-Tx would be extremely useful to identify patients at risk of DGF and prevent this common complication, for example delaying start of CNI [7].

Ideally, biomarkers predicting DGF should be available either before KTx, in the donor, or immediately after it, in the recipient. The first option is especially interesting in the current era of increasingly higher risk ECDs [8], as accurate tools to assess kidney quality are needed to help allocate them to the most adequate recipient, or even discard them when considered unsuitable [9].

A lot of potential biomarkers of DGF have been studied and some of them have already been validated in independent cohorts (Table 2 and Table 3). Some biomarkers have been analyzed in donor’s biological fluids or in the graft (e.g., preservation fluid) before KTx, whereas most of them were studied in the recipient after KTx.

### 2.1. Donor-Related Biomarkers

Donor-related biomarkers can be measured in donor biological fluids, in graft preservation fluid, or in the perfusate of machine-perfused kidneys.

#### 2.1.1. Donor Biological Fluids

Elevated donor plasma mitochondrial DNA levels independently predicted DGF and correlated with 1-year graft survival in a cohort of DCD [10].

Following organ procurement, the role of innate immune system, such as Complement in IRI, has been extensively investigated. By generating effector molecules (C4b, C4d, C3b, iC3b, C3dg, and C3d) and anaphylatoxins (C3a, C5a), Complement can recruit granulocytes, monocytes, and other inflammatory cells to the site of ischemic injury and regulate activation of tubular epithelial cells and pericytes within the kidney. In addition, Complement factors can directly damage renal parenchymal cells by inducing tubular apoptosis, endothelial-to-mesenchymal transition (EndMT), pericytes-to-mesenchymal transition, and accelerated senescence [28,29,30,31]. EndMT deeply modifies endothelial cells, which acquires a mesenchymal phenotype and new properties, such as capacity to produce extracellular matrix (EM) and induce fibrosis. Biomarkers of EndMT have been the focus of recent research in different KTx areas and will be discussed in detail in following sections on recipient-related DGF biomarkers (Section 2.2.4) and on chronic rejection-IFTA within the setting of CAD (Section 4.1.3).

Consistently, donor urinary C5a levels were independently associated with recipient post-transplant DGF, providing a potential rationale for complement-blocking therapies to prevent DGF in high risk patients [11].

#### 2.1.2. Graft Preservation Fluid

Other studies have recently focused on analysis of potential biomarkers within graft preservation fluids, especially during hypothermic machine perfusion, with the rationale that their concentration may reflect organ viability and correlate with post-transplant renal function [12].

Cell-free microRNAs (miRNAs) show promise as biomarkers in several KTx settings. These are short non-coding RNAs that play a pivotal role in regulation of gene expression through epigenetic, transcriptional, and post-transcriptional mechanisms. They can be isolated, quantified and profiled by multiple platforms which can also characterize their target genes [32]. They have been studied in graft preservation fluid and proposed as viability biomarkers (miR-486-5p, miR-144-3p, miR-142-5p, and miR-144-5p) [12]; however, only miR-505-3p has been demonstrated to be an independent predictor of DGF in DCD grafts with high accuracy (AUC = 0.83) and was confirmed in a validation cohort [13].

Of note, a significant percentage of miRNAs do not circulate free but are carried by EVs that have been detected in preservation fluid. These structures contain both donor-derived RNAs and selected miRNA which could be associated with graft function during the first seven post-operative days [33].

General features of EVs and their role as biomarkers of DGF will be discussed in more detail in the following paragraphs.

#### 2.1.3. Perfusate of Machine-Perfused Kidneys

Proteomic analysis of perfusate from machine cold perfusion of graft was compared between different types of donor kidneys. LDH, neutrophil gelatinase-associated lipocalin (NGAL), and matrix metalloproteinase-2 levels were highest in DCD kidneys, followed by DBD and living-donor (LD) kidneys. Other molecules, such as periredoxin-2 and α-1 antitripsin, were also significantly different across the three groups, probably reflecting different degrees of IRI [14,15]. Exosomal mRNA for NGAL and NGAL concentration in the perfusate of machine-perfused kidneys were associated with DGF also in another study [15]. The α and π iso-enzymes of glutathione S-transferase (GST) levels, measured from perfusate solution at the start and the end of machine perfusion, were analyzed in 428 KTx recipients. While levels of both iso-enzymes significantly increased during this procedure, only πGST levels at the end of machine perfusion were independently associated with DGF [16].

All the above-mentioned molecules represent potential biomarkers and therapeutic targets that may be useful in the setting of DGF, but still need to be validated.

### 2.2. Recipient-Related Biomarkers

#### 2.2.1. Furosemide Stress Test

Furosemide stress test (FST) is a simple test to predict DGF in the post-transplant period. FST non-responsive patients (urine volume < 350 cc after 4 h of Furosemide infusion) are at risk of developing DGF in the following days [17].

#### 2.2.2. miRNAs

miRNAs, which we already analyzed as donor-derived biomarkers, have been the focus of several studies also in KTx recipients, representing both a biomarker and a potential therapeutic target [34,35,36]. MiR 182-5p and miR-21-3p in recipient’s serum and urine correlated with DGF in one study [18]. MiR 146a-5p has been studied in renal tissue and peripheral blood during DGF. It was significantly increased in renal biopsy of patients with DGF as compared to stable recipients and those with AR and a similar trend was found in peripheral blood samples [19].

A urinary panel of six miRNAs (miR-9; miR-10a; miR-21; miR-29a; miR-221; miR-429) was consistently elevated in the first urine passed after Tx and in urine samples collected daily across the following five post-operative days in patients who developed DGF (ROC AUC = 0.94). This panel was validated in an independent cohort [20].

In experimental IRI studies in mice, the expression of miR-139-5p in renal tissues of the IRI group was 40% lower than that of the sham-operated one. A set of candidate genes involved in regeneration and repair of kidney tissue, EM degradation and inflammation was also shown to be markedly overexpressed in this setting and may provide new biomarkers in the future [37].

#### 2.2.3. Neutrophil Gelatinase-Associated Lipocalin (NGAL) and Other Biomarkers

NGAL has been the focus of many studies as a tubular injury biomarker for early prediction of DGF in KTx recipients. It has been studied both in graft perfusion fluid and in recipient’s blood and urine.

Increased release from ischemia-injured tubular cells has been proved to discriminate patients at risk for AKI. Blood NGAL (bNGAL)—performed on serum/plasma—and urine NGAL (uNGAL) were shown to predict DGF in the early post-operative period, whereas its meaning as a perfusion fluid biomarker has already been discussed [15,21].

In one study on 50 KTx recipients from ECD, bNGAL levels at day 1 were significantly higher in the DGF group; of interest, NGAL accurately discriminated between slow and immediate graft function even within the non-DGF group. Furthermore, bNGAL levels preceded decrease in serum creatinine and allowed earlier TAC introduction in a “sequential” immunosuppressive protocol, shortening CNI-free window as compared to standard, creatinine-based management. Thus, bNGAL may help avoid unnecessary CNI underexposure in patients in which renal function is about to recover. The same study also shed light on NGAL function as a growth factor for tubular epithelial cells. In vitro, either hypoxia or TAC exposure induced its release from tubular epithelial cells and NGAL stimulated their regeneration after IRI and acute nephrotoxicity through an autocrine loop. However, chronic tubular stimulation by NGAL also appeared to promote epithelial-to-mesenchymal transition (EMT) and progression toward CKD. This pathological process will be discussed in detail in the following Section 4.1.3 concerning mechanisms of chronic rejection and IFTA. Overall these data suggest that NGAL levels might even predict a maladaptive repair with increased risk of progression from DGF to chronic loss of graft function [38].

Consistently, a more recent study prospectively assessed dynamic profile of bNGAL and uNGAL in 170 consecutive recipients within 7 days of Tx and found that their level on post-operative day 2 could accurately predict DGF. Multivariate analyses revealed donor age, serum and urinary NGAL were each independently associated with DGF (*p* < 0.001) [21].

A metanalysis first demonstrated that elevated serum and urine NGAL levels can predict DGF and 1-year graft function [22]; a second, more recent one, including 1036 patients from 14 studies, confirmed that both bNGAL—performed on serum/plasma—and uNGAL were robust biomarkers for DGF (AUC 0.91 and 0.95, respectively), with superior predictive value of bNGAL over uNGAL [23].

Of interest, urine NGAL post-operative modification in the first 24 hours were associated not only with DGF but also with worse renal outcomes at 2 years in terms of graft function and survival in LD KTx [24].

Several other biomarkers have been proposed in the setting of DGF.

A urinary tissue inhibitor of metalloproteinases-2 (TIMP-2), a validated biomarker for AKI, was reported to predict the occurrence and duration of DGF in DCD KTx recipients [39].

In a transcriptomic study on IRI mice, Corin was one of the most downregulated among more than 2200 differentially expressed genes and protein level of renal Corin was markedly reduced in IRI. Consistently, also plasma Corin concentrations were reduced in a small sample of recipients with DGF as compared to uncomplicated KTx recipients [25].

Expression of Toll-like (TLR-4) expression on circulating monocytes was reported to be lower in DGF patients and associated with poor graft function at follow-up [26].

An increase in serum Amylase (>20%), especially if associated with increased Resistive Index (>0.7) predicted a higher incidence of DGF, longer hospital stay, and worse renal function at discharge in another study [27].

#### 2.2.4. BioMarkers of EndMT

In a recent study biomarkers of partial microvasculature EndMT (Fascin and Vimentin) and of tubular EMT (Vimentin) were analyzed with immunoistochemistry in renal biopsies performed in early post-transplant due to DGF, showing ATN lesions. Extent of ATN was correlated with short and long-term (2 year) graft dysfunction only in the presence of partial EndMT (pEndMT) biomarkers expression, suggesting that early endothelial cell activation can identify patients at risk of incomplete recovery after DGF [28]. EndMt will be discussed in detail in the following Section 4.1.3 concerning mechanisms of chronic rejection and IFTA.

#### 2.2.5. EVs

EVs is a general term which includes membrane structures of different size, released by cells after fusion of endosomes with the plasma membrane (exosomes), shed from plasma membrane (microvesicles), or released during apoptosis (apoptotic bodies). EVs are then taken up by neighboring or distant target cells (paracrine or endocrine effect) [40] and mediate a wide range of physiological and pathological processes, including renal disease [41]. EVs also exert pleiotropic, immunomodulatory roles in KTx [42]. Their bioactive cargo includes graft antigens, costimulatory/inhibitory molecules, cytokines, growth factors and, as discussed before, functional miRNAs that modulate expression of recipient cell target genes. Recent studies dissected this complex content, suggesting that some of these molecules may be potential biomarkers of DGF, paralleling recovery of renal and endothelial function. Even though initial evidence on dynamics of circulating EVs after KTx needs to be confirmed [43], this area of research appears to be promising.

Plasma and urinary EVs investigated as possible biomarkers of DGF in KTx are outlined in Table 3 [44,45,46,47,48,49,50].

**Table 3 ijms-21-05404-t003:** Extracellular vescicles (EVs) as potential biomarkers of DGF.

Type of EV	Main Features	Author
Plasma Endothelial EVs	EVs level and their procoagulant activity progressively decrease after KTx, paralleling renal function recovery	Al-Massarani G et al. [44,45]
Plasma Endothelial and platelet EVs	Endothelial and platelet EVs size and level progressively decrease after KTx, paralleling renal function recovery	Martins S et al. [46]
Urinary EVs	NGAL expression in urinary EVs correlated with DGF	Alvarez S et al. [47]
Urinary CD 133+ EVs	Decreased level in recipients with DGF and vascular damage	Dimuccio V et al. [48]
Acquaporin-1 containing EVs	Decreased urinary Acquaporin-1-containing EVs in DGF	Sonoda H et al. [49] Asvapromtada S et al. [50]

## 3. AR

Potential biomarkers of acute antibody-mediated rejection (ABMR) and T-cell mediated rejection (TCMR) are reported in detail in Table 4, Table 5 and Table 6.

### 3.1. Transcriptomic Studies

#### 3.1.1. Urine and Peripheral Blood Transcriptomics

The CTOT 04 trial has analyzed mRNA transcripts in urinary sediment cells and identified a three-gene signature (CD3ε mRNA, CXCL10 mRNA, and 18S rRNA) predicting TCMR up to 20 days before biopsy-proven diagnosis [51]. A more recent study by the same group analyzed gene expression in urinary cells and renal biopsies during AR and identified unique and shared gene signatures associated with biological pathways involved in TCMR and ABMR. Furthermore, they demonstrated the enrichment of biopsy gene signature in urinary cells and of immune cell types in urine compared with renal tissue. These findings support the hypothesis that urine gene expression patterns can reflect and even amplify ongoing renal tissue immune pathways and may help diagnose rejection and monitor its dynamics [85]. This is consistent with evidence from previous studies suggesting that graft can sort renal tissue infiltrating cells in urine as an in vivo flow cytometer [86].

Several studies have tried to identify a peripheral blood gene expression signature to diagnose subclinical AR at an early stage.

Christakoudi S et al. analyzed expression of 22 literature-based genes in peripheral blood samples of patients from Kidney Allograft Immune Biomarkers of Rejection Episodes (KALIBRE) study and identified a seven-gene TCMR-signature (IFN-γ, IP-10, ITGA4, MARCH8, RORc, SEMA7A, WDR40A) which allowed diagnosis of AR 7 weeks before renal biopsy and correlated with response to therapy [52].

Zhang W et al. focused on patients with subclinical TCMR (protocol biopsy at third month) in KTx recipients from Genomics of Chronic Allograft Rejection (GoCAR) study [87] and identified a 17-gene peripheral blood signature which characterized ongoing subclinical TCMR and predicted an increased risk of clinical TCMR at 24 months and decreased graft survival [53].

A peripheral blood mRNA assay based on eight genes (CXCL-10, FCGR1A, FCGR1B, GBP1, GBP4, IL15, KLRC1, TIMP1) was developed in a multicenter, prospective study and correlated with histological features of acute and chronic ABMR (microvascular inflammation, transplant glomerulopathy) but not of TCMR. Diagnostic accuracy was high (ROC AUC 79.9% *p* < 0.0001), even in the setting of stable graft function [54].

A blood molecular biomarker based on multiple gene expression signatures was designed to distinguish “immunological quiescence” from subclinical AR in a multicenter study (CTOT-08). This correlated with clinical (AR, renal function) and histological outcomes (IFTA) and with de novo DSA. This biomarker was validated with surveillance biopsies data and proved to be especially useful in ruling out subclinical rejection (NPP: 78–88%) [55].

In a multi-center “Assessment of Acute Rejection in renal Transplant (AART)” study, peripheral blood transcriptome analysis identified a 17-gene signature called “Kidney Solid Organ Response test” (kSORT), which predicted both TCMR and ABMR up to 3 months before histological diagnosis in an independent prospective cohort. This tool is characterized by a high accuracy in predicting AR, especially when compared with performance of other biomarkers in the same setting (AUC = 0.94; sensitivity: 83%, specificity: 90.6%; PPV: 93,2%) [57,88].

kSORT has also been used in association with an IFNγ Elispot in the ESCAPE study, resulting in a higher PPP (AUC > 0.85) for subclinical TCMR and ABMR [74].

Peripheral blood transcriptomic analysis allowed to build a classification model capable of discriminating ABMR from accommodation in ABO-incompatible kidney transplants [89].

A blood test (TruGraf v1) has been developed to study a set of microarray-based gene expression in order to discriminate patients with a stable graft and immunological quiescence (“Transplant Excellence”) from those with renal dysfunction or AR. This tool was proposed as a tool to avoid unnecessary surveillance biopsies on the basis of high accuracy in detecting AR (74%) and high NPP (90%) [90]. The value of serial TruGraf testing to confirm immunoquiescence and avoid surveillance biopsies has been confirmed in a recent study [91].

#### 3.1.2. Renal Tissue Transcriptomics

Sigdel T et al. analyzed tissue expression of selected 19 target genes, including those previously identified in tissue common rejection module (tCRM). Interestingly, they employed RNA extracted from archival fresh frozen paraffin-embedded renal biopsy tissue. Eight genes were related to specific cellular infiltrates, whereas the others reflected a “graft inflammation score” based on tCRM. This set of genes allowed to distinguish biopsies of stable grafts from those of recipients with AR and even borderline inflammation [56].

Molecular patterns such as upregulation of intrarenal complement regulatory genes discriminate accommodation from subclinical antibody-mediated rejection in AB0-incompatible KTx [92].

An intra-graft mRNA transcriptomic landscape of TCMR has been outlined through computational analysis and has shown an increased expression of innate immunity genes, such as genes for pattern recognition receptors, and a decreased expression of calcineurin, suggesting inadequate immunosuppression, as compared to stable graft [61].

Real time central molecular assessment of changes in mRNA expression in graft tissue through microarrays is the basis of “molecular microscope diagnostic system”, which predicted risk of AR and graft failure with greater precision than conventional biopsy. Pathogenesis-based transcripts sets (PBTs) which segregate together and characterize different processes (e.g., IGF-gamma expression, T cell infiltrates), were employed to define “classifiers” which predict molecular phenotype, quantifying its likelihood with a score. Of importance, this approach has been validated in several independent cohorts [93,94].

Other studies have shown that endothelial associated transcripts (ENDATs) in biopsies of DSA-positive patients can reveal ABMR even in the absence of C4d positivity [58] and that ABMR-related endothelial genes RNA transcripts are expressed before histological onset of lesions, allowing excellent identification (AUC = 0.92) and potentially early, preemptive treatment of rejection [59].

Finally, single-cell transcriptomics can comprehensively describe cell types and states in a human kidney biopsy and was employed to analyze immune response in mixed rejection: 16 distinct cell types were identified, including different sub-clusters of activated endothelial cells [95]. This cell-based approach may provide a wealth of new biomarkers for ABMR in the future [96].

### 3.2. Complement-Related Biomarkers

Complement system is deeply involved in ABMR and can therefore provide potential biomarkers related to this process.

The C4d deposition has been considered the gold standard for ABMR diagnosis for several years, indicating activation of Classical pathway of Complement; however, all Complement pathways have been proved to be involved in ABMR, leading to recruitment and activation of leukocytes such as Natural Killer cells, monocytes/macrophages, and lymphocytes [97].

Bobka S. et al. also demonstrated an increased Complement activation in pre-transplant biopsies from diabetic, hypertensive, or smoking donors, suggesting a predictive value of Complement activation in donor biopsies for later outcome [98]. Expression of these Complement components at time of diagnosis of ABMR was associated with higher serum creatinine and more severe morphological changes. As further evidence, C5 blockade prevented ABMR and stabilized long-term renal function.

In addition, EVs shed by endothelial cell expressing C4d (CD144^+^ C4d^+^) are increased in ABMR and correlate with its severity and response to treatment [99] and plasma levels of complement activation fragments C4a and Ba are increased in ABMR [60]. Single nucleotide polymorphisms (SNP) of complement C3 gene have also been found to correlate with ABMR [100]. Upregulation of intrarenal complement regulatory genes and complement transcripts in peripheral blood of ABO-incompatible KTx has already been discussed in “Transcriptomic Studies” [92]. Altogether, these data support the use of Complement Factors as potential biomarkers in ABMR.

### 3.3. Urinary and Serum Chemokines

IFN-γ induced urinary C-X-C motif chemokine ligand 9 (CXCL9) and 10 (CXCL10) chemokines are associated with Th-1 immune response and involved in T cell recruitment in inflammatory processes. They are promising as biomarkers for TCMR and ABMR [62,66].

Low levels are associated with immunological quiescence, as shown by their very high NPP, which makes them an ideal tool to rule out rejection, including subclinical ones, and to identify transplant recipients at low immunological risk [63]. This was especially evident for CXCL 9 (CTOT-01 study), which was associated with acute TCMR within the first year.

However, a subsequent study with a longer follow-up (CTOT-17) showed that changes in eGFR between 3 or 6 months and 24 months better predicted 5-year graft loss than CXCL-9 measurement [64].

Association of urinary CXCL10-to-creatinine ratio with DSA improved identification of ABMR and prediction of graft loss. In a recent study, higher blood and urine levels of both CXCL9 and CXCL10 were found in ABMR, but urinary CXCL9 was the most accurate biomarker of rejection (AUC of ROC: 0.77) and—if measured in combination with immunodominant DSA mean fluorescence intensity (MFI)—it allowed a net reclassification increase of 73% compared to DSA MFI alone [65].

Interestingly, even CXCL9 and CXCL10 baseline recipient’s serum levels assessed before KTx may predict AR [101,102].

Additionally, urinary CXCR3—the receptor for CXCL9 and CXCL10, expressed on activated T-lymphocytes—was shown to detect subclinical inflammation and correlate with evolution towards chronic damage; of interest, its level decreased after immunosuppression intensification [103].

In another study, serum concentration of chemokine CXCL13, a B lymphocyte chemoattractant, was significantly higher in TCMR than in stable graft and in borderline rejection; furthermore, a marked increase (>5-fold) was found in patients developing AR within first post-transplant week and correlated with entity of B cell infiltration in renal biopsy. A similar correlation was found in a mouse model of TCMR, indicating that CXCL13 serum levels may be a marker of B cell-involvement in TCMR, identifying a severe subset of this type of rejection [104].

On the whole, growing evidence points to a role of urinary and serum chemokines as biomarkers of both types of AR.

### 3.4. Other Potential Urinary Biomarkers

Other urinary molecules have been proposed as markers of AR.

High urinary π-GST values at postoperative day 1 discriminated AR (sensitivity, 100%; specificity, 66.6%) as well as between DGF from normal-functioning grafts (sensitivity, 100%; specificity, 62.6%). Similarly, α-GST values > 33.97 ng/mg uCrea identified AR, with a lower sensitivity (77.7%) but optimal specificity (100%) [105].

Urinary untargeted metabolomic profiling led to identification of a panel of five potential biomarkers (guanidoacetil acid, methylimidazolacetic acid, dopamine, 4-guanidobutyric acid, and L-tryptophan), which discriminated between TCMR and stable graft (ROC curve AUC: sensitivity 90%; specificity 84.6%) [106].

### 3.5. dd-cfDNA

Small fragments of cell-free DNA, released from graft cells into the recipient circulation due to cell death or injury, have been proposed as biomarkers of AR.

While dd-cf DNA represents on average 0.34% of total cf-DNA in plasma of stable KTx recipients, levels are increased during AR and, to a lesser extent, acute pyelonephritis and ATN.

A kinetic pilot study of dd-cfDNA after Tx showed high median level in the immediate post-Tx hours (around 20%), rapidly decreasing on the first day (around 5%) and then stabilizing below 1% [107].

In the DART (Diagnosing Active Rejection in Kidney Transplant Recipients) trial, Bloom RD et al. first reported higher dd-cfDNA levels in patients with acute (TCMR Banff > IB and ABMR) and chronic active rejection and identified a 1% threshold to discriminate these patients from stable ones. This test was characterized by elevated NPP (84%) and a lower PPP (61%), suggesting that <1% percentage of dd-cfDNA could be used to rule out rejection, especially ABMR. Coexistence of DSA increased PPP to 85%. Furthermore, dd-cfDNA levels not only increased before changes in serum creatinine but also decreased after rejection treatment, suggesting that longitudinal monitoring of this biomarker could be useful after a rejection episode, possibly limiting need for surveillance biopsies [67].

In a subsequent work, Huang E et al. [68] demonstrated that a lower dd-cfDNA threshold of 0.74% could reliably identify ABMR—but not TCMR—in a group of immunologically high-risk patients undergoing indication biopsies, increasing NPP to 100%.

However, other authors suggested comparable performance of dd-cfDNA in diagnosing ABMR and TCMR, using a different quantification methodology [69].

Absolute quantification of dd-cfDNA (copies/mL) showed superior performance in discriminating BPAR as compared to dd-cfDNA percentage and also seemed to identify a subset of patients with inadequate Tacrolimus levels and subclinical immunological damage in a prospective observational study [70].

Of interest, dd-cfDNA diagnostic capacity for ABMR appears to improve when applied to DSA-positive recipients, suggesting a preferential employment in monitoring highly sensitized patients [71].

Determination of dd-cfDNA can be unreliable in case of recent (within 1 month) whole blood transfusion and falsely positive within 24 h of a renal biopsy; it should also not be employed to monitor a second KTx as release from previous graft could alter its levels. Falsely positive results can also occur in the case of ATN and acute pyelonephritis and type of donor also affects levels (higher levels in cadaveric vs. LD), probably reflecting difference in degree of initial ischemic damage and inflammation [72].

Despite these issues, dd-cfDNA remains a promising biomarker and it has been proposed as a surrogate diagnostic ABMR criterion in DSA-negative forms [108].

Furthermore, recent studies suggest that dd-cfDNA determination could also have a broader meaning beyond AR diagnosis, reflecting graft injury and consequently exerting a negative impact on several long-term outcomes [109]. Of interest, a multicentric study on patients with initial TCMR (TCMR 1A and borderline lesions) showed that dd-cfDNA levels above 0.5% were effective in stratifying risk of eGFR decline, de novo DSA development and further AR episodes [110]. Consistently, emerging evidence indicates that levels of dd-cfDNA increase before onset of de novo DSA (both HLA-DSA and non-HLA DSA) and eGFR decline [73], suggesting that dd-cf DNA itself is immunogenic and can trigger subclinical inflammation, initiating an immune response [75].

Urinary levels of cell-free mitochondrial DNA during early post-transplant phase have also been reported to correlate with AR, DGF and short-term renal function [76].

### 3.6. Allogenic Circulating B-Cell and T-Cell Assays

Peripheral circulating donor HLA-specific memory B cells quantified by enzyme-linked immunospot (ELISPOT) [77] and serum B-cell activating factor level on post-operative day 7 [78] both predicted ABMR, especially in DSA-positive recipients.

Pre-transplant T cell alloreactivity can be assessed with a donor-specific IFN-γ ELISPOT which measures IFN-γ release by recipient T cells in response to donor antigens. IFN-γ ELISPOT intensity appears to correlate with development of subclinical TCMR, ABMR, and DSA [79,80].

### 3.7. Peripheral Blood miRNAs

General features and meaning of miRNAs have already been dealt with in the paragraph on DGF biomarkers. A panel of five peripheral blood miRNAs—miR-15b, miR-16, miR-103a, miR-106A, miR107—was shown to improve sensitivity of diagnosis of vascular TCMR [81].

### 3.8. Immune Cells Biomarkers

Peri-transplant soluble CD30 (sCD30), a marker of activated T-cell mediated immunity, has been reported to predict early AR [82].

A recent metanalysis on 18 studies (1453 total patients) has confirmed a strong association between sCD30 and AR, especially for KTx from deceased donors [83].

CD154-positive T cytotoxic memory cells were associated with acute TCMR and its histological severity in a small cohort of KTx recipients receiving steroid-free TAC after alemtuzumab induction [84].

Pre-transplant, baseline levels of CD200 (a protein belonging to immunoglobulin superfamily) and CD200R1 (its myeloid-cell specific receptor, which mediates inhibitory signals) have been analyzed in a monocentric cohort of 125 KTx recipients; an increased pre-transplant CD200R1/CD200 ratio identified recipients at increased risk of AR and worse renal function at the 3rd and 6th month after KTx [111].

Additionally, pre-transplant expression of CD45RC on circulating CD8+ T lymphocytes predicted AR (mainly TCMR); a percentage of CD8+CD45RC T cells above 58.4% was independently associated with a 4-fold increase in the risk of AR [112].

### 3.9. Non-HLA DSA

Donor human leukocyte antigen (HLA)-specific antibodies were initially identified as a major cause of ABMR. This type of DSA has been extensively studied and represents an established, “traditional” biomarker of ABMR, which is beyond the scope of this review [113,114].

In more recent years, preformed and de novo non-HLA specific DSA targeting G-protein coupled receptors expressed on graft glomerular endothelium have been the focus of intense research, as they may account for a significant proportion of HLA-DSA negative acute and chronic ABMR [115,116,117]. They include a wide range of autoantibodies against different antigens, all of which represent potential biomarkers for ABMR [118] (Table 5).

Antibodies against type 1 receptor for Angiotensin 2 (AT1R) and Endothelin type A receptor (ETAR) are the most studied non-HLA, activating antibodies and appear to exert their effect either alone or in synergy with DSA. After binding to their receptors, these autoantibodies phenotypically modify and activate endothelial cell by triggering different intracellular pathways. They probably represent a bridge between allo- and autoimmunity within rejection, as these two components can interact and amplify one another [119]. Pre-transplant antibodies against AT1R and ETAR may identify a subset of patients at higher risk for acute and chronic rejection and graft loss, independent of HLA-directed alloimmune response [120,121], possibly even in a setting of low-immunological risk such as KTx from LD [122,123,124]. Pre-transplant antibodies against AT1R have also been associated with more severe microvascular inflammation histological lesions as compared to negative patients [125].

Anti-vimentin antibodies detected before KTx, probably reflecting previous endothelial damage occurred during hemodialysis, have also been associated with graft dysfunction [126].

Anti-Perlecan/LG3 antibodies are produced as a consequence of Perlecan release from injured endothelial cells [127]. They are highly prevalent in hypersensitized patients [128] and have been associated with acute ABMR, DGF, and reduced long term survival [129,130].

Anti-endothelial cell antibodies (AECA), which include a wide range of autoantibodies against several surface antigens, may also prove to be a source of rejection biomarkers [131,132].

In general, AECA have been associated with acute and chronic rejection and with early graft dysfunction in different types of solid organ transplant, including heart and kidney. De novo AECA seem to be more strongly associated with ABMR than preformed ones [133].

Identification of their target antigens is complex, and their precise meaning must still be elucidated for most of them, as they could represent biomarkers of past vascular injury or, on the contrary, be active contributors to microvascular inflammation [134].

However, some specific types of AECA have already been clinically characterized and show promise as biomarkers of endothelial injury. Their antigenic targets are Endoglin, Fms-like tyrosine kinase-3 ligand (FLT3-L), EGF-like repeats and discoidin I-like domains 3 (EDIL-3), and intercellular adhesion molecule 4 (ICAM-4), all involved in endothelial cell activation and leukocyte adhesion and margination. AECA have been associated with de novo DSA, ABMR, and early transplant glomerulopathy [131]. More recently, also anti-keratin-1 (KRT-1) antibodies were found to be associated with an increased risk of AR [132].

Finally, development of antibodies directed against tissue-specific self-antigens, such as Fibronectin (FN) and Collagen type IV (Col IV), increases the risk of AR in pancreas-kidney transplantation (PKT) [135] and transplant glomerulopathy in KTx [136]. These autoantibodies probably reflect breakdown of tolerance towards self-antigens, as suggested by detection of self-Ag-specific IFN-γ and IL-17 secreting T-cells in the same patients. Therefore, they could provide a biomarker of a tissue-specific autoimmune component of rejection.

In the near future, improved identification and characterization of non-HLA DSAs may help better classification of ABMR subphenotypes and provide diagnostic and prognostic biomarkers and potentially even indication for preemptive specific therapies in this subset of patients [124].

**Table 5 ijms-21-05404-t005:** Non-HLA DSA as a potential biomarker for antibody-mediated rejection (ABMR).

Biomarker	Main Features	Author
Anti-AT1R	Pre-transplant levels associated with, acute and chronic ABMR, severity of microvascular inflammation, graft dysfunction, and graft loss	Dragun D et al. [119] Philogene MC Hum Imm 2019 [120] Sas-Strozik et al. [121] Shinae Y et al. [122] DF Pinelli et al. [123] MA Lim et al. [125]
Anti-ETAR	Pre-transplant levels associated with acute and chronic ABMR graft dysfunction and graft loss	Philogene MC et al. Hum Imm 2019 [120] Shinae Y et al. [122] DF Pinelli et al. [123] Jackson AM et al. [131]
Anti-Vimentin	Pre-transplant levels associated with graft dysfunction	Dyvanian T et al. [126]
Anti-Perlecan	Highly prevalent in hypersensitized patients. Pre-transplant levels associated with increased risk of DGF, acute ABMR, and reduced long-term function	Dieudè M et al. [127] Riesco L et al. [128] Padet L et al. [129] Yang B et al. [130]
AECA	They include a variety of antibodies against endothelial antigens (Endoglin, FLT-3, EDIL-3, ICAM-4, KTR-1) and correlate with increased risk of ABMR	Jackson AM et al. [131] Guo X et al. [132] Sanchez Zapardiel E et al. [133]
Anti-FN and Col-IV	De novo development increases risk of AR (PKT) and transplant glomerulopathy (KTx)	Angaswamy N et al. [135] Gunasekeran M et al. [136]

### 3.10. Other Biomarkers

Another potential biomarker is serum N-glycan determination, performed at days 1 and 7 post-Tx and integrated in a clinical score (including age, gender, and immunological risk factors). A higher sum of scores at days 1 and 7 (>0.5) predicted graft rejection (AUC = 0.87) and correlated with long-term rejection-free survival in a cohort of LD Tx recipients [137].

Heat shock protein 90 (HSP-90), a molecular chaperon protein released into serum by damaged cells, was found to be significantly elevated in plasma of KTx with AR as compared to stable graft and other pathological conditions (chronic rejection, CNI nephrotoxicity, Polyomavirus nephropathy) and returned to baseline after immunosuppressive treatment [2,138].

Heparan Sulfate plasma levels are increased in TCMR compared to stable graft, due to release from EM during graft T-cell infiltration [2,139].

Many other urinary and plasmatic proteins could be potential biomarkers of rejection but deserve to be further studied: among these, C-C motif chemokine ligand 2 (CCL2), NGAL, IL-18, cystatin C, KIM-1, T-cell immunoglobulin and mucine domains-containing protein 3 (TIM3), alpha-1 antitrypsin (A1AT), alpha-2 antiplasmin (A2AP), serum amyloid A (SAA), and apolipoprotein CIII (APOC3) [2,140,141].

### 3.11. EVs

General features and meaning of EVs have already been dealt with in the paragraph on DGF biomarkers.

EVs represent a versatile tool given the huge variety of mediators included in their cargo. Therefore, potential applications of plasma and urinary EVS as biomarkers have also been studied in AR, as outlined in Table 6. In some studies, EVs levels have been considered as biomarkers themselves [99], whereas a set of specific molecules included in their cargo proved to be a potential biomarker of AR in others [142,143,144,145,146].

**Table 6 ijms-21-05404-t006:** EVs as potential biomarkers of AR.

Type of EV	Type of Rejection	Main Features	Author
Plasma C4d+CD144+ endothelial EVs	ABMR	Levels correlate with ABMR presence and severity and decrease after successful treatment	Tower C et al. [99]
Plasma endothelial EVs	ABMR	A combination score based on 4 mRNA transcripts overexpressed in EVs of patients with ABMR predicts imminent rejection in HLA- sensitized patients	Zhang H et al. [142]
Plasma endothelial EVs	ABMR	Levels increase in ABMR and decrease after treatment in the early post-transplant; however, they are also influenced by renal function recovery	Qamri Z et al. [143]
Urinary EVs	TCMR	A total of 11 protein enriched in urinary EV in patients with TCMR	Sigdel T et al. [144]
Urinary EVs	TCMR	A total of 17 protein enriched in urinary EV in patients with TCMR; Tetraspanin-1 and Hemopexin proposed as biomarkers	Lim J et al. [145]
Urinary EVs	TCMR	High levels of CD3 + EVs released by T-cell in urine are strongly associated with TCMR	Park J et al. [146]

## 4. Chronic Allograft Dysfunction (CAD)

Chronic allograft dysfunction is the main cause of long-term graft loss [147].

Different entities can be accounted for this picture, with chronic ABMR (cABMR) playing a predominant role in most cases [148].

However, other components can be represented by CNI nephrotoxicity, PVAN, de novo or relapsing glomerulonephritis. Many studies have focused on biomarkers for late graft dysfunction as a global entity, while others have tried to identify specific biomarkers to dissect each of these components.

In general, defining specific biomarkers for CAD is difficult, because molecular fingerprints of acute and chronic rejection are overlapping, partly reflecting similar mechanisms. Some authors propose a “threshold effect”, with AR developing when intensity of alterations is high and chronic rejection expressing a less important degree of alterations [2]. For example, Complement is not only involved in ABMR, as described in a previous paragraph, but also plays a pivotal role as mediator of tubular senescence [28,30,149] and interstitial fibrosis, premature aging phenomena that characterize progression to chronic damage [150]. C3a, C5a, and the terminal C5b-9 complex can each amplify damage during CKD progression. Anaphylatoxins bind to their specific receptors inducing pro-inflammatory and fibrogenic activity on tubular and endothelial cells [151,152], pericytes [153], and resident fibroblasts, whereas C5b-9 complex can regulate production of pro-fibrotic and pro-inflammatory cytokines [97]. Collectively, these data indicate that uncontrolled Complement activation may result in maladaptive tissue repair with irreversible development of renal fibrosis and aging. Identification of biomarkers of CAD is therefore challenging due to coexistence of acute and chronic processes, but it would be extremely useful for a differential diagnosis [154].

### 4.1. Chronic Rejection and IFTA

Potential biomarkers for chronic rejection and IFTA are outlined in Table 7. IFTA is found in around 25% of 1-year biopsies and correlates with decreased graft survival when histological evidence of inflammation is present.

#### 4.1.1. Transcriptomic Studies

Growing evidence of highly shared deregulated gene pathways between IFTA and AR suggests a common immunological etiology in most cases of late CAD [154].

Recent studies have focused on upregulation of genes involved in IFTA. Inflammation in IFTA areas (“inflammatory IFTA”, i-IFTA) has been identified as pivotal element in prompting development of chronic renal damage, further underlying the relationship between chronic, subclinical immunological activity and irreversible fibrosis [161,162].

Several transcriptomic studies have shed light on specific genes and miRNAs involved in fibrotic evolution of chronic rejection.

In a study by Mas V. et al. an upregulation of genes related to fibrosis (TGFβ), extracellular matrix deposition, and immune response was found [155].

In the already quoted CTOT-04 trial, Lee J. R. et al. identified a four-gene urinary signature (mRNA for vimentin, NKCC2, E-cadherin, and 18S rRNA) which predicted IFTA [86].

In the study of Genomics of Chronic Allograft Rejection (GoCAR), renal biopsy transcriptome expression analysis identified a set of 13 genes which independently predicted development of CAD at the 12th month, despite normal histology at the 3rd month after KTx, in more than 200 prospectively followed patients with stable graft function. This multicenter study was validated in two independent cohorts and first raised hope that allograft injury may be detected before it becomes clinically evident [87].

Halloran et al. employed the “molecular microscope” approach (already discussed in the paragraph on AR) and demonstrated a progressively higher prevalence of IFTA lesions over time and its association with transcripts related to rejection and glomerulonephritis in late biopsies. This suggests a continuing, active tissue response rather than autonomous fibrogenesis and that early abrogation of the immunological process may be critical to block this evolution and preserve long-term graft function [93,161].

Another transcriptomic study employed an 85-gene signature related to IFTA and employed it to test targeted new anti-fibrotic drugs [156].

#### 4.1.2. miRNAs

miRNAs, which we already analyzed as candidate biomarkers in the setting of DGF and AR, are also opening new perspectives in this setting. Recent studies have proposed sets of urinary and renal biopsy miRNAs as prognostic biomarkers of IFTA and CAD [163].

Aberrant urinary mi-R21 and miR200b expression was associated with IFTA and CAD [157].

Plasma circulating levels of miR-150, miR-192, miR-200b, and miR-423-3p were significantly different between patients with IFTA and those with stable renal Tx and accurately identified IFTA (AUC = 0.87; sensitivity = 78%; specificity = 91%) [158].

In another study, plasma expression of miR-21, miR-142-3p, and miR-155 were upregulated in IFTA and mi-R 21 levels were positively correlated with eGFR [159].

On the contrary, miR-145-5p expression in blood cells was significantly downregulated in IFTA and could discriminate it from many other active lesions, such as TCMR, ABMR, borderline-rejection, and from a condition of stable graft function [160].

Another area of active research is that of epigenetic modifications of immunity genes on progression to IFTA: epigenetic mechanisms such as hypomethylation could directly enhance their expression and also indirectly modulate it by regulating miRNAs [164].

#### 4.1.3. Biomarkers of EMT and EndMT

IF is determined by massive deposition of EM, which is mainly produced by activated myofibroblasts probably derived from several cell types, especially renal tubular cells, through EMT.

This process, promoted by several factors such as oxidative stress and mitochondrial dysfunction due to IRI, deeply alters epithelial cell properties, determining loss of polarity and cell–cell adhesion and assumption of a mesenchymal phenotype, characterized by markedly increased production of EM [165].

More recently, activated myofibroblasts have been shown to arise also from renal endothelial cells through a similar process, EndMT, already mentioned in the section on DGF [166].

Both EMT and EndMT lead to abnormal production of EM and consequently play a key role in the pathogenesis of allograft IFTA [161]. Several histological and urinary EMT biomarkers have been proposed (Table 8), whereas more recent, initial evidence on potential EndMT biomarkers in KTx is available. Biomarkers of both processes will be analyzed.

(a)Biomarkers of EMT


*Histological biomarkers*


In a recent study, renal expression of CD45, vimentin (VIM), and periostin (POSTN) correlated with iIFTA and POSTN was the strongest predictor of graft loss. Of interest, its expression was inversely correlated with 25(OH)VitD levels, suggesting that these might influence graft fibrosis [167].

Smad ubiquitination regulatory factor 1 (Smurf1) is part of Smurf1/Akt/mTOR/P70S6K signaling pathway, activated by TNF-α and involved in EMT. Of interest, Bortezomib blunted progression of EMT and IF by inhibiting TNF-α production and consequently expression of Smurf1, suggesting that this could be an EMT biomarker with diagnostic and therapeutic value [168].

Tubular expression of VIM and β-catenin, biomarkers of EMT, in protocol biopsy performed 3 months after KTx, was an independent risk factor for IFTA and eGFR decline up to 4 years post-transplant in CsA-treated recipients [169].

Finally, an interesting area of research is that of cellular senescence. This is associated with an inflammatory, “senescence-associated secretory phenotype” (SASP) which is tightly connected to EMT and CAD. Senescence markers (e.g., p16^INK4a^) could therefore be considered as potential surrogate biomarkers of EMT [170].


*Urinary biomarkers*


An interesting non-invasive biomarker of EMT is the ratio between VIM and CD45 relative to uroplakin 1a (UPK) urinary mRNA, which has been shown to correlate with intensity of VIM renal expression measured with immunostaining in per-protocol renal biopsies [171].

Other studies adopting a whole transcriptomic analysis approach identified specific urinary transcriptomic patterns associated with pEMT. Unbiased pathway analysis revealed that these patterns expressed increased inflammation and reduced metabolic functions, suggesting that they may be effective to detect subclinical immune response leading to EMT and graft fibrosis [172].

(b)Biomarkers of EndMT

Three biomarkers of EndMT, fascin1, vimentin, and heat shock protein 47, were strongly expressed in endothelial cells of peritubular capillaries in ABMR as compared to stable patients and predicted late graft dysfunction (up to 4 years since ABMR diagnosis) better than histological lesions. These results suggest that they may be reliable in identifying persistent endothelial activation and evolution towards cABMR [173].

In vitro and in vivo experimental studies demonstrated that EndMT may promote IF by targeting the TGF-β/Smad and Akt/mTOR/p70S6K signaling pathways, indicating that components of these pathways may be a potential source of EndMT biomarkers [174].

Finally, E Glover et al. analyzed evidence of miRNAs regulation of EndMT from experimental studies and their potential impact on kidney and other solid organ allograft dysfunction in a recent review. However, clinical studies in humans are needed to confirm their role as EndMT biomarkers [175].

### 4.2. Chronic CNI Nephrotoxicity

Some other studies identified potential specific biomarkers for chronic CNI nephrotoxicity, which are outlined in Table 9.

Chronic ischemia due to the vasoconstrictive effect of CNI triggers an alteration in expression of proteins involved in pro-inflammatory response and oxidative stress; however, the renal histology of chronic CNI nephrotoxicity is not peculiar (it may in fact merely determine IFTA) and this hampers efforts to identify specific biomarkers [176].

A metabolomic study compared urine from healthy subjects and KTx recipients with biopsy-proven chronic TAC nephrotoxicity and proposed symmetric dimethylarginine and serine as marker of this type of kidney injury (ROC analysis AUC of 0.95 and 0.81, respectively) [177].

uNGAL was proved to correlate with duration of CsA therapy in children with CNI nephrotoxicity [178].

A SNP in the FK-506-binding protein (FKBP), rs6041749 C variant, appeared to enhance FKBP1A gene transcription compared to the T variant and was associated with an increased risk of CAD in a Chinese cohort of TAC-treated KTx recipients, although with an unclear mechanism [179].

Other studies in rat models have reported increased urinary levels of TNAα, LIM-1, and FN in the early phase of CsA nephrotoxicity and late increases of urinary Osteopontin and TGF-β in chronic nephrotoxicity [180].

Decreased expression of Slc12a3 and KS-WNK1, leading to impaired sodium transport in distal tubules and chronic activation of renin-angiotensin system, was associated with CsA and TAC nephrotoxicity in another rat model [181]. Potential biomarkers identified in the last two experimental studies need to be validated in humans.

**Table 9 ijms-21-05404-t009:** Potential biomarkers for chronic calcineurin-inhibitor (CNI) nephrotoxicity.

Biomarker	Main Features	Author
Urinary symmetric dimethylarginine and serine	Highly accurate for CNI nephrotoxicity (AUC of 0.95 and 0.81, respectively)	Xia T et al. [177]
uNGAL	It correlates with duration of CsA therapy in children with CNI nephrotoxicity	Gacka E et al. [178]
Genetic polymorphism of FK-506-binding protein, rs6041749 C variant	It enhances FKBP1A gene transcription and is associated with an increased risk of CAD in TAC-treated KTx recipients	Wu Z et al. [179]
Increased urinary TNAα, LIM-1, FN Osteopontin, and TGF-β	These markers correlate with different stages of CsA nephrotoxicity in rat models	Carlos C et al. [180]
Decreased renal expression of Slc12a3 and KS-WNK1	These markers correlate with different stages of CNI nephrotoxicity in rat models	Cui Y et al. [181]

### 4.3. PVAN

Potential biomarkers for PVAN, an important cause of CAD [182], are outlined in Table 10.

Urinary exosomal bkv-miR-B1-5p and bkv-miR-B1-5p/miR-16, two miRNAs encoded by PVAN, have both demonstrated very high discriminative capacity for this complication (ROC AUC 0.98 for each) as compared with that of commonly used surrogate biomarkers, such as plasma viral load [183].

Urinary CXCL10 has been associated with subclinical inflammation within the tubule-interstitial and peritubular capillary spaces and correlated with Polyomavirus viremia [184].

A single nucleotide polymorphism (SNP) of IL28B (C/T polymorphism rs12979860) was associated with presence of PVAN, discriminating these patients from those with viremia without any renal involvement [185].

The search for renal tissue transcriptomic biomarkers of PVAN has not provided any solid result so far. Overlap in pathogenetic mechanisms and gene expression between PVAN and non-viral forms of allograft injury, such as TCMR and iIFTA, makes it difficult to identify peculiar molecular signatures [186].

## 5. Current Limits and Perspectives of Biomarkers in Renal Transplant

Advances in high-throughput technologies have been providing an avalanche of new potential biomarkers over the last decade. However, in general, their application in clinical practice is currently being restrained by several drawbacks. Most available biomarkers do not meet ideal requirements outlined in Table 11 and certainly require further validation through multicenter studies, as single-center discovery step often inflates their value [187].

Most important, their role and cost-effectiveness should be assessed in prospective randomized trials designed to compare them with standard KTx management with traditional diagnostic tools.

Despite these limits, biomarkers represent the cornerstone of precision medicine, which aims at integrating traditional clinical information and tailoring medical care to select the best treatment for an individual patient [5]. This new frontier will probably deeply change the way we monitor KTx and manage its complications.

Renal biopsy, the traditional gold standard for assessing graft dysfunction, is usually triggered by a change in serum creatinine and/or proteinuria and has a limited diagnostic power for initial injury, when histological changes are minimal or equivocal [3]. By contrast, an ideal biomarker (or a set of biomarkers) should lead to an earlier and more objective diagnosis (Table 11) making it possible to pre-emptively treat histological initial lesions long before they become irreversible, or even before they become visible with traditional tools, marking patterns of molecular alterations which predate histological injury (“molecular rejection”). Biomarkers could decrease the need for renal biopsy to detect subclinical disease (e.g., protocol biopsies) and even substitute for it when contraindicated. Furthermore, while current new potential biomarkers in KTx mainly have a diagnostic/prognostic meaning, the area of monitoring, pharmacodynamic/response, and safety biomarkers (Table 1) is substantially unexplored in this setting and could help us improve long-term management of allograft dysfunction (e.g., follow-up of patients after BPAR, with repeated, non-invasive monitoring biomarkers to rule out persistence of ongoing subclinical rejection; assessment of etiology and degree of activity/chronicity in CAD).

Particularly interesting perspectives are immunological risk stratification and identification of low-risk, or even tolerant patients.

Peripheral blood gene expression tests such TruGraf [91] or kSORT [57] have already become commercial and appear accurate in identifying a state of “immunological quiescence” in stable recipients; due to their high NPP they could allow to rule out ongoing subclinical rejection through serial monitoring, as an alternative to surveillance biopsies, and guide immunosuppression minimization in fragile patients at low immunological risk [188].

A further step forward would be to identify biomarkers of operational tolerance, a rare condition characterized by maintenance of stable renal function without any immunosuppressive therapy.

Tolerant patients seem to be depicted by increased expression of B cell associated genes in the blood and urine and by a peculiar B cell repertoire, enriched in naive and transitional B cells. Of interest, this pattern appears to be associated with better long-term graft function [189] and potential biomarkers of this process are beginning to emerge. For example, TCL1A, an oncogene expressed in immature naive and transitional B cells, and promoting their survival, has been associated with immunosuppressive properties of this lymphocyte sub-population and seems to be upregulated in stable, rejection-free KTx recipients [190].

Of interest, Newell et al. identified a B-cell signature formed by a set of three genes which correlated with increased expression of CD20 mRNAs (FoxP3, CD20, CD3, perforin) in urinary sediment of tolerant patients compared to healthy controls (all of them) and to stable KTx (only CD20) [191], whereas Danger et al. showed that a composite score based on a 20-gene signature peripheral blood cells could accurately discriminate operationally tolerant recipients from stable ones, independent of immunosuppressive therapy [192]. All these approaches need to be validated, but they may pave the way for the identification of tolerance biomarkers, with important implications on management of immunosuppressive therapy [193]. The state-of-the-art of this family of biomarkers was recently analyzed in several reviews [2,194,195] and is beyond the scope of this work.

At the other end of the spectrum, biomarkers could be preferably employed to monitor high-immunological risk patients (e.g., sensitized, DSA-positive recipients). Testing biomarkers in this subset helps increase PPP due to a higher a priori risk of AR. A combination of different biomarkers can also increase diagnostic accuracy; for example, association of kSORT with IFNγ ELISPOT improves predictive power for subclinical TCMR and ABMR [74].

Another intriguing perspective is the application of artificial intelligence (AI) models which allows computational analysis and interpretation of large-scale molecular data generation by exploiting machine learning algorithms and neural networks [196,197]. For example, classifiers like artificial neural networks, support vector machines and Bayesian inference have already been employed in pilot studies to screen KTx recipients requiring renal biopsy [198] and AI has proved useful to improve estimation of TAC Area Under the Concentration Over Time Curve [199].

“Molecular microscope” is another important example application of AI to renal tissue transcriptomic analysis [93,94].

In another recent work an unsupervised learning method integrating a wide range of parameters (clinical functional, immunologic, and histologic) was applied to a large cohort of KTx recipients and allowed to classify five transplant glomerulopathy archetypes, each associated with a different allograft 5-year graft survival (ranging from 88% to 22%) [200].

These studies suggest that progress in AI can significantly contribute to a completely new, more accurate disease nosology, integrating complex sets of biomarkers of different nature (from clinical data to molecular aspects) for a subtle characterization of traditional entities.

## 6. Conclusions

Development of Omics technology and expanding knowledge of new tools, such as EVs and dd-cfDNA, has led to an increased availability of a wide range of new potential biomarkers, which may be applied to all key settings of early and late graft dysfunction. Non-invasive biomarkers measured in urine or blood appear promising in providing very early diagnosis of pathological processes, such as subclinical AR, or in stratifying risk of DGF or of rejection, potentially reducing need for surveillance biopsies to monitor low-risk recipients. Tissue biomarkers have also proved effective in integrating traditional histology, leading to improved disease nosology and more accurate prognosis. Tolerance biomarkers and progress in AI are opening new frontiers, which may revolutionize transplant medicine.

Although larger, multi-center validation studies are needed before combination of biomarkers can be widely implemented in the clinic, the transplant physician should rise to the challenge of becoming familiar with this new landscape, in order to start taking advantage of the various facets of its huge potential.

## Figures and Tables

**Figure 1 ijms-21-05404-f001:**
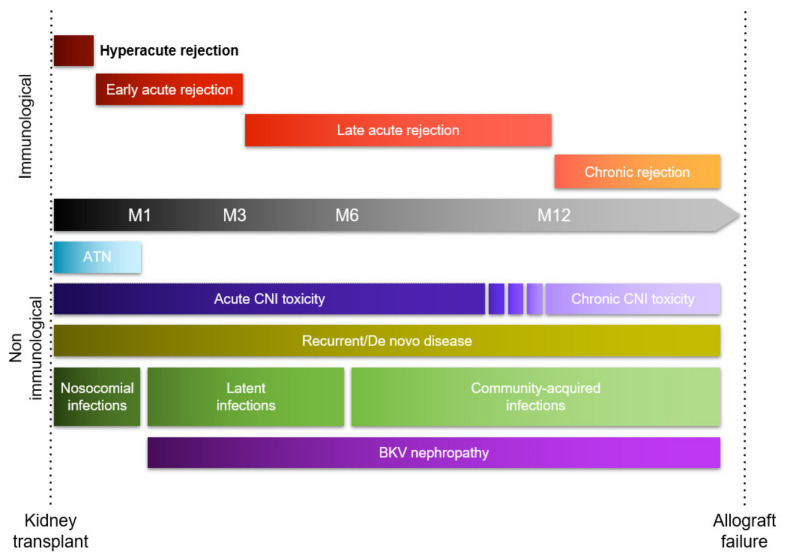
Timeline of early and late causes of graft dysfunction.

**Table 1 ijms-21-05404-t001:** Biomarkers categories and their meaning in renal transplant.

Type of Biomarker	Meaning in Renal Transplant
Susceptibility or risk biomarker	It estimates the risk of developing a condition (e.g., AR) in a stable graft without any clinical sign of dysfunction
Diagnostic biomarker	It identifies patients with a disease or a subset of it (e.g., AR type)
Prognostic biomarker	It estimates the likelihood of a clinical event or of disease progression, staging severity of disease (e.g., severe rejection with risk of graft loss)
Predictive biomarker	It estimates the likelihood of achieving a favorable response from a therapy (e.g., Eculizumab for complement-fixing DSA)
Monitoring biomarker	It is serially measured in order to detect a change in evolution of disease or signs of drug toxicity, or to detect exposure to immunosuppressive drugs (e.g., TAC levels)
Pharmacodynamic/response biomarker	It verifies that a biological response has occurred after a drug exposure (e.g., DSA MFI after treatment of ABMR)
Safety biomarker	It estimates presence and severity of drug-related toxicity (e.g., CNI nephrotoxicity)

**Table 2 ijms-21-05404-t002:** Potential biomarkers for DGF.

Biomarker	Source	Main Features	Author
Mitochondrial DNA	Donor plasma	It predicts DGF in DCD donors	Han F. et al. [10]
Complement C5a	Donor urine	It predicts DGF	Schroppel B. et al. [11]
miRNA	Graft preservation fluid	Several miRNAs proposed as biomarkers of DGF; miR-505-3p validated in DCD grafts	Gomez-Dos-Santos V. et al. [12] Roest H. et al. [13]
LDH, NGAL and MMP-2	Perfusate of machine-perfused kidneys	Different levels according to type of donor (DCD vs. DBD vs. LD), reflecting degree of IRI	Moser M. et al. [14]
Exosomal mRNA for NGAL and NGAL	Perfusate of machine-perfused kidneys	They predict DGF	Cappuccilli M. et al. [15]
πGST	Perfusate of machine-perfused kidneys	It predicts DGF	Hall I. et al. [16]
Furosemide stress test	---	Clinical test: non-responsive patients are at increased risk of DGF in the following days	Udomkarnjananun S. et al. [17]
miR182-5p, miR-21-3p	Recipient’s serum and urine	They predict DGF	Wilflingseder J. et al. [18]
miR146a-5p	Recipient’s peripheral blood and renal tissue	Increased in both DGF and AR	Milhoransa P. et al. [19]
miR-9, miR-10a, miR-21, miR-29a, miR-221, miR-429	Recipient’s urine (first 5 days after KTx)	This panel predicts DGF (validated in an independent cohort)	Khalid U. et al. [20]
NGAL	Recipient’s serum/plasma and urine (first days after KTx)	Both bNGAL and uNGAL predict DGF and 1-year graft function, but bNGAL is more accurate. Urine NGAL predicts DGF also in KTx from LD.	Cappuccilli M. et al. [15] Maier H. et al. [21] Ramirez-Sandoval J. et al. [22] Li Y. et al. [23] Sahraei Z. et al. [24]
Corin	Recipient’s plasma	It is reduced in DGF	Hu X. et al. [25]
TLR-4 surface expression	Recipient’s circulating monocytes	It is reduced in DGF and associated with poor graft function at follow-up	Zmonarski S. et al. [26]
Amylase	Recipient’s serum	It increases in DGF	Comai G. et al. [27]
Fascin and Vimentin	Graft biopsy in recipient	Expression of these EndMT biomarkers on microvasculature correlated with long-term graft function after DGF	Xu-Dubois Y-C. et al. [28]

**Table 4 ijms-21-05404-t004:** Potential biomarkers for acute rejection (AR).

Biomarker	Type of Rejection	Main Features	Author
Three-gene signature (CTOT 04 study)	TCMR	It increases up to 20 days before histological diagnosis	Suthanthiran M et al. [51]
Seven-gene signature (KALIBRE study)	TCMR	It increases 7 weeks before histological diagnosis and decreased after treatment	Christakoudi S et al. [52]
Seventeen-gene signature (GoCAR study)	TCMR	It identifies subclinical TCMR and correlates with long-term graft survival	Zhang W et al. [53]
Eight-gene signature	ABMR	It correlates with histological features of acute and chronic ABMR	Van Loon E et al. [54]
Panel of gene signature (CTOT 08 study)	TCMR and ABMR	It correlates with clinical and histological outcomes and with de novo DSA; useful to identify immunologically quiescent patients	Friedewald J et al. [55]
Nineteen-gene signature	TCMR and ABMR	It includes TCMR genes. Analysis performed on RNA extracted from archival fresh frozen paraffin-embedded renal biopsy tissue.	Sigdel T et al. [56]
kSORT (AART study)	TCMR and ABMR	Rejection predicted 3 months before histological diagnosis in 64% of patients with stable graft function.	Roedder S et al. [57] Zhang W et al. [53]
ENDATs	ABMR	Analysis of endothelial transcripts predicts ABMR with excellent accuracy (AUC = 0.92).	Sis B et al. [58] Adam B et al. [59]
Complement fragments	ABMR	Levels correlate with ABMR	Stites E et al. [60]
Innate immunity genes	TCMR	Unbiased transcriptome analysis identifies increased expression of innate immune system genes	Mueller F et al. [61]
CXCL9	TCMR and ABMR	High NPP (99.3%): low levels at 6 months predict low risk of rejection until 24 months. Highly accurate for ABMR diagnosis when associated with DSA.	Hricik D et al. [62] Rabant M et al. [63] Faddoul G et al. [64] Mühlbacher J et al. [65]
CXCL10	ABMR and mixed	High NPP (99%). It predicts rejection at 1 month post-KTx in stable graft.	Rabant M et al. [66]
dd-cfDNA	ABMR and TCMR	Due to elevated negative NPP, it could help rule out especially ABMR and play a role for surveillance after a rejection episode or in sensitized patients	Bloom R et al. [67,68,69,70,71,72]
Allogenic circulating B- and T-cell assays	ABMR and TCMR	Useful to predict subclinical forms of rejection and DSA	Hricik D et al. [73] Crespo E et al. [74] Gorbacheva V et al. [75]
Peripheral blood miRNAs	TCMR	miR-15b, miR-16, miR-103a, miR-106A, miR107 predict vascular TCMR	Matz M et al. [76]
Peritransplant soluble CD30 (sCD30)	TCMR	Strong association between sCD30 and TCMR	Trailin A et al. [77] Mirzakhani M et al. [78]
CD154-positive T cytotoxic memory cells	TCMR	Association with TCMR and its histological severity in steroid-free regimen	Ashokkumar C et al. [79]
CD 200 and CD200R1	TCMR and ABMR	Increased pre-transplant CD200R1/CD200 ratio identifies recipients at increased risk of AR and worse renal function	Oweira H et al. [80]
CD45RC	TCMR	Pre-transplant expression of CD45RC on circulating CD8+ T predicts AR	Lemerle M et al. [81]
N-glycan	ABMR and TCMR	N-glycan levels (integrated within a clinical score) predict rejection-free survival in KTx from LD	Soma O et al. [82]
HSP-90	ABMR and TCMR	It discriminates AR from other causes of graft dysfunction	Maehana T et al. [83]
Heparan Sulfate	TCMR	It predicts DGF	Barbas A et al. [84]

**Table 7 ijms-21-05404-t007:** Potential biomarkers for chronic rejection and interstitial fibrosis-tubular atrophy (IFTA).

Biomarker	Main Features	Author
Set of genes related to fibrosis (i.e., TGFβ), extracellular matrix deposition and immune response	Upregulated in IFTA	Mas V et al. [155]
4-gene urinary signature (mRNA for vimentin, NKCC2, E-cadherin, and 18S rRNA)	It predicts evolution of chronic rejection towards IFTA	Lee J. et al. [86]
13-gene renal tissue signature (GoCAR study)	It predicts CAD at the 12th month even with normal histology at the 3rd month	O’Connell P. et al. [87]
85-gene renal tissue signature	Associated with IFTA	Li L. et al. [156]
Urinary mi-R21 and mi-R200b	Increased expression predicts IFTA and CAD	Zununi V. et al. [157]
Plasmatic miR-150, miR-192, miR-200b, and miR-423-3p	Highly accurate in identifying IFTA (AUC = 0.87; sensitivity = 78%; specificity = 91%)	Zununi V. et al. [158]
Plasmatic miR-21, miR-142-3p, miR-155, and mi-R 21	Upregulated in IFTA; mi-R 21 correlates with GFR	Zununi V. et al. [159]
miR-145-5p expression in blood cells	Downregulated in IFTA; It can discriminate it from acute and borderline rejection	Matz M. et al. [160]

**Table 8 ijms-21-05404-t008:** Potential biomarkers for epithelial-to-mesenchymal transition (EMT).

Biomarker	Main Features	Author
CD45, VIM, and POSTN	They correlate to each other and with iIFTA and graft loss	Alfieri C et al. [167]
Smurf 1	It is included in a pathway involved in EMT. Its inhibition by Bortezomib may mediate its anti-fibrotic effect.	Zhou J et al. [168]
VIM and β-catenin	Tubular expression correlates with IFTA and long-term eGFR decline	Hazzan M et al. [169]
Senescence biomarkers (e.g., p16INK4a)	They mark SASP, an inflammatory phenotype connected to EMT	Sosa Pena DPM et al. [170].
VIM and CD45 relative to UPK mRNA	This ratio based on urinary mRNAs correlates with VIM expression in renal tissue and may detect EMT and early graft fibrogenesis	Mezni I et al. [171]
Urinary transcriptomic patterns	They are associated with pEMT and subclinical graft injury	Galichon P et al. [172]

**Table 10 ijms-21-05404-t010:** Potential biomarkers for Polyomavirus-associated nephropathy (PVAN).

Biomarker	Main Features	Author
Urinary exosomal bkv-miR-B1-5p and bkv-miR-B1-5p/miR-16	Excellent diagnostic accuracy for PVAN	Kim M et al. [183]
Urinary CXCL10	Associated with subclinical tubule-interstitial inflammation and viremia	Ho J et al. [184]
IL28B SNP C/T (rs12979860)	Associated with presence of PVAN in viremic patients	Dvir R et al. [185]

**Table 11 ijms-21-05404-t011:** Features of an ideal biomarker for kidney transplant (KTx) [1,2,187].

Biomarker Features	Comment
Non-invasive and easy to measure	Urine and blood biomarkers are easily available and can be serially measured, whereas renal tissue biomarkers require renal biopsy with inherent invasiveness and limits. Urine and blood biomarkers may be used when renal biopsy is contraindicated or reduce the need for repeated surveillance biopsies.
Short turn-around time	Results should be available within a time frame which allows rapid, potentially pre-emptive intervention (e.g., diagnosis of subclinical AR)
Easy to interpret	Results should be easy to interpret, and threshold values should be established to help transplant physician in clinical practice
Reproducible and standardized	Results should be validated in multiple independent cohorts with different features (e.g., elderly, or highly sensitized KTx recipients, or different ethnicity) and assay standardization of analytical process performed in order to minimize inter-laboratory and inter-platform variability
Accuracy (sensitivity and specificity)	Biomarker levels should strictly reflect a single specific pathological process, without being influenced by other causes of kidney damage (e.g., AR vs. CNI nephrotoxicity or vs. infections)
Good prognostic performance (PPV and NPV)	Acceptable PPP and NPP. In general, new biomarkers should be preferably tested in subsets of patients at different immunological risk, rather than on the transplant population as a whole, in order to improve their statistical performance (e.g., higher a priori chance of AR in highly sensitized KTx recipients improves PPV compared to standard recipients).
Proof of cause	Reduction of a biomarker level correlates with an improvement in the underlying pathological process assessed with current gold-standard (histological examination with renal biopsy)
Cost-effective	Results should improve clinical management and consequently impact long-term outcomes and related economic aspects, justifying biomarker costs (e.g., a biomarker which detects subclinical AR could improve treatment, prolong graft survival and reduce costs)

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
