# Peer review of "Recent Advances on Biomarkers of Early and Late Kidney Graft Dysfunction"

_ijms, 2020, doi:10.3390/ijms21155404_

Round 1

Reviewer 1 Report

Overall, this is important review in the field of solid organ transplantation.

The investigators, however, should additional discuss the use of important markers of allograft injury including

  • Donor-Derived Cell-Free DNA for Assessment of Allograft Rejection and Injury Status (The Use of Donor-Derived Cell-Free DNA for Assessment of Allograft Rejection and Injury Status. J Clin Med. 2020;9(5):1480. Published 2020 May 14. doi:10.3390/jcm9051480)
  • Furthermore, non-HLA autoantibodies detection should also be discussed including: 

    AT1R

    ETAR

    Vimentin

    Perlecan

    Endoglin

    FLT3 ligand

    EDIL3

    ICAM4

    Fibronectin

    Collagen type 4
  • Markers of Endothelial-To-Mesenchymal Transition has been mentioned but should be additionally discussed in more details with dedicated figure.

Author Response

Answers to Reviewer 1

  1. We additionally discussed significance and potential applications of Donor-Derived Cell-Free DNA (dd-cfDNA) as biomarker of allograft rejection and injury (Paragraph 3.5, page 12; modifications highlighted in yellow) adding 5 new References of recent studies, including the one quoted by the Reviewer;
  1. We inserted a new Paragraph (3.9, page 13) on non-HLA DSA as potential biomarkers of rejection, as requested, and 25 relevant References including all recent studies on this topic;
  1. We inserted a new Paragraph (4.1.3, page 17) on markers of epithelial-to-mesenchymal transition (EMT) and endothelial-to-mesenchymal transition (EndMT), which we had just mentioned in the first version. The Paragraph was inserted in the section dedicated to chronic rejection-IFTA in the setting of Chronic Allograft Dysfunction, with a dedicated Table and 11 relevant References; in addition, markers of EndoMT are also discussed in the setting of DGF-ischemia reperfusion injury, where a recent study has been introduced. We propose adding a Table dedicated to EMT markers, instead of a Figure, because we think this could be more suitable to clarify and summarize the text, also taking into account the heterogeneity of biomarkers we discuss.  

Reviewer 2 Report

The authors reviewed biomarkers of kidney graft dysfunction. This is the comprehensive review including various factors including donor, ischemia, rejection and drug toxicity. There are some miss spelling and font (ex. line 282 and line 637) and the authors should revise them.

Author Response

Answers to Reviewer 2

We corrected misspellings and font errors and thouroughly revised the text, as indicated (modifications highlighted in yellow)

Round 2

Reviewer 1 Report

all of my comments have been addressed